# Vertical Profiles of Particle Number Size Distribution and Variation Characteristics at the Eastern Slope of the Tibetan Plateau

**Chenyang Shu [1], Langfeng Zhu [2], Yinshan Yang [3], Xingbing Zhao [4], Xingwen Jiang [4], Hancheng Hu [5], Dongyang Pu [2], Mengqi Liu [2] and Hao Wu [2],***

[1] Plateau Atmospheres and Environment Key Laboratory of Sichuan Province & School of Atmospheric Sciences, Chengdu University of Information Technology, Chengdu 610225, China; 3200107011@stu.cuit.edu.cn

[2] Key Laboratory of China Meteorological Administration Atmospheric Sounding, School of Electrical Engineering, Chengdu University of Information Technology, Chengdu 610225, China

[3] Faculty of Geographical Science, Beijing Normal University, Beijing 100875, China

[4] Institute of Plateau Meteorology, China Meteorological Administration, Chengdu 610072, China

[5] College of Optoelectronic Engineering, Chengdu University of Information Technology, Chengdu 610225, China

* Correspondence: wuhao@cuit.edu.cn

**Abstract:** An unmanned aerial vehicle (UAV) observation platform obtained the first vertical profiles of particle number size distribution (PNSD) from 7 to 16 July 2022 on the eastern slope of the Tibetan Plateau (ESTP). The results were from two flanks at the Chuni (CN) and Tianquan (TQ) sites, which are alongside a mountain (Mt. Erlang). The observations revealed a significant negative correlation between the planetary boundary layer height (PBLH) and the particle number concentration (PNC), and the correlation coefficient was −0.19. During the morning, the rise in the PBLH at the CN and TQ sites caused decreases of 16.43% and 58.76%, respectively, in the PNC. Three distinct profile characteristics were classified: Type I, the explosive growth of fine particles with a size range of 130–272 nm under conditions of low humidity, strong wind shear, and northerly winds; Type II, the process of particles with a size range of 130–272 nm showing hygroscopic growth into larger particles (e.g., 226–272 nm) under high humidity conditions (RH > 85%), with a maximum vertical change rate of about −1653 # cm$^{-3}$ km$^{-1}$ for $N_{130-272}$ and about 3098 # cm$^{-3}$ km$^{-1}$ for $N_{272-570}$; and Type III, in which during the occurrence of a surface low-pressure center and an 850 hPa low-vortex circulation in the Sichuan Basin, polluting air masses originating from urban agglomeration were transported to the ESTP region, resulting in an observed increase in the PNC below 600 nm. Overall, this study sheds light on the various factors affecting the vertical profiles of PNSD in the ESTP region, including regional transport, meteorological conditions, and particle growth processes, helping us to further understand the various features of the aerosol and atmospheric physical character in this key region.

**Keywords:** vertical profiles; particle number size distribution; eastern slope of the Tibetan Plateau

## 1. Introduction

Aerosols are colloids composed of fine solid particles and liquid droplets suspended in the atmosphere [1], which are key constituents of the Earth's atmosphere system [2]. They play an important role in the climate and weather changes, and have complicated relationships with extreme weather and atmospheric pollution [3–5]. Aerosols play a significant role in the Earth's energy budget and climate change by directly altering solar radiation and indirectly interacting with clouds' optical, microphysical, and microphysical properties [2,4,6]. To evaluate the effects of aerosols on the atmosphere, knowledge of the size distribution of particle numbers is needed [7]. In the real atmospheric environment,

aerosols are commonly characterized by four size distributions: the nucleation mode (diameter particle (Dp) Dp < 30 nm), the Aitken mode (30 nm < Dp < 100 nm), the accumulation mode (100 nm < Dp < 1 mm), and the coarse mode (Dp > 1 mm) [8–11].

The particle number size distribution (PNSD) is one of the most crucial properties for characterizing aerosols, which reflects the formation mechanisms, dynamic processes, and chemical transformations experienced by aerosol particles of different sources, physical properties, and chemical compositions in the atmosphere [8,12,13].

Nevertheless, the factors that influence aerosol size distributions are still not well understood. To clarify the effects of aerosols, more knowledge of the size-resolved particle number emissions and their sources is needed. The increase in the number of aerosols in the nucleation and Aitken mode is primarily influenced by the occurrence of new particle formation (NPF) events [7,14–16]. Except for NPF, under conditions conducive to the formation and growth of new particles, such as during NPF events or in the presence of specific atmospheric processes [17–19], the explosive growth of fine particles typically occurs, which denotes a rapid and significant increase in the number or concentration of particles in a specific size range. The occurrence of explosive growth can result in a sudden augmentation of particles, with significant implications for air quality, climate, and human health. Understanding the underlying mechanisms and factors contributing to this explosive particle growth is vital for the accurate assessment of particle impacts [20]. Furthermore, these newly formed particles can potentially be activated as cloud condensation nuclei (CCN) after hours of growth [21]. Research indicates that approximately half of the global CCN may originate from NPF events [19,22–24]. Furthermore, traffic emissions and culinary activities are two of the significant factors contributing to the increase of aerosol particles in the nucleation and Aitken modes [7,25]. The emissions from light-duty gasoline vehicles are primarily characterized by a geometric mean diameter (GMD) below 20 nm, whereas emissions from heavy-duty diesel vehicles are predominantly associated with a GMD around 100 nm [12,26–28], while the particle size range associated with culinary emissions falls within a range of approximately 20–200 nm [18,29–31]. In addition, the particle number concentration in the accumulation mode can be affected by the aged aerosols transported regionally [32–34].

Additionally, meteorological factors modify the growth mode and growth rate of particles, thereby affecting the PNSD. Observational studies have revealed that the evolution of the planetary boundary layer (PBL) and its vertical mixing play a crucial role in determining the diffusion of aerosols and their gaseous precursors [35,36]. This intricate process is influenced by a combination of thermodynamic factors, including temperature, humidity, and stability, as well as turbulent dynamics such as eddy diffusion and entrainment. These interconnected processes have a significant impact on the occurrence of NPF events and the PNSD [37,38]. In northern China, northwest or northeast winds have been found to promote the formation of nucleation mode particles, while easterly winds facilitate the growth of these particles [14,39]. In the context of Delhi, India, coarse mode particles are predominant before the onset of the monsoon season, whereas accumulation mode particles become dominant afterward [40].

The transitional zone between the Sichuan Basin (SCB) and the Tibetan Plateau is known as the ESTP, characterized by significant variations in topography as the altitude increases. This region is influenced by the East Asian monsoon, Indian monsoon, and thermodynamics forcing circulation on the Tibetan Plateau, which result in distinct flux transport and convergence patterns [6,41]. Furthermore, the SCB, characterized by intricate topography and a densely populated area, has become one of the most severely polluted regions in China [42,43]. In contrast, the Tibetan Plateau, situated at a high altitude with minimal human activity, is renowned for its pristine and uncontaminated environment, which provides an ideal observation site for studying the background atmospheric composition. However, the vertical profile of PNSD in the ESTP region has received limited attention in previous research. The unique thermodynamic conditions over the plateau also contribute to the vertical stratification of aerosols. Thermodynamically forced circulation,

such as that caused by temperature inversions and diurnal heating, can cause aerosols to accumulate or disperse at specific altitudes. Regional transport plays a crucial role in shaping the vertical profiles of PNSD in the ESTP. The transport of aerosols from the SCB and surrounding cities can significantly influence the physical and chemical properties of aerosols in the ESTP. During long-range transport, aerosols can undergo aging processes, undergoing chemical and physical changes that can alter their properties, including light absorption efficiency. Given the complexity of these interactions, it is essential to conduct further research on the vertical profiles of PNSD in the ESTP region. Understanding these processes will provide valuable insights into the atmospheric dynamics, climate, and air quality of this ecologically sensitive and geographically important area [44,45].

The characterization of aerosol distribution in the vicinity of the PBL has been extensively investigated using various methods such as ground-based LiDAR, meteorological towers, manned aircraft, tethered balloons, and satellites [46–48]. However, the need for higher spatiotemporal resolution and data on microphysical characteristics has led to the increasing utilization of unmanned aerial vehicles (UAVs) in atmospheric measurements. UAVs offer advantages in terms of flexibility, frequent flights, ease of deployment, and the ability to carry various sensors and detection equipment [49,50]. Therefore, numerous recent studies have employed UAVs to measure aerosol vertical profiles in urban areas [51–53]. Yet, in regions with complex terrain or land cover, only a handful of investigations have utilized UAVs, such as the field observations in the Taklimakan Desert by Zhou et al. [54]. In general, there is still a paucity of research on PNSD in regions characterized by complex underlying surfaces. This limitation constrains our comprehension of the factors that influence PNSD in the ESTP region.

In the Tibetan Plateau and its surrounding areas, there has been limited research on the aerosol particle size distribution profiles in and upon the boundary layer, and often inadequate temporal resolution is a common issue [55–57]. To address this gap, this study conducted PNSD vertical observations at two sites in the ETSP region, Chuni, and Tianquan, using UAVs. For detailed information regarding the data and methods employed, please refer to Section 2. Section 3 presents a comprehensive description of the aerosol particle size distribution profiles during the summer in the ESTP region and provides a discussion of their influencing factors. Finally, Section 4 offers a summary and further discussion of the findings.

## 2. Data and Methods

### 2.1. Observation Site

The field campaign was carried out in the ESTP region, which represents the transitional zone from the SCB to the Tibetan Plateau and is characterized by its high-mountain gorge topography (Figure 1). From 7–11 December 2022, a UAV took off at Chuni (CN site, 29.81°N, 102.20°E), which is nestled within a valley in the western foothills of Erlang Mountain, sitting at an altitude of 1289 m. Similarly, from 12–16 December 2022, a UAV took off at Tianquan (TQ site, 29.92°N, 102.38°E), which is located in the eastern foothills of Erlang Mountain, approximately 20 km away from the CN site, and at an altitude of 1417 m. Both observation sites are situated in sparsely populated areas where the influence of human activities and emissions is considerably less pronounced compared to that in urban areas.

### 2.2. Instrumentation and Data

#### 2.2.1. Vertical Profile Measurement

A hexacopter UAV (model Hi-Drone M600, XH230E, Junpan Electromechanical Ltd. Co., Nanjing, China) was utilized as a platform to carry observation instruments. The UAV, equipped with a 326,000 mAh battery, had a maximum carrying capacity of 8 kg and can hover for up to 80 min without additional load, for 60 min with a 2 kg load, and 32 min with an 8 kg load. During the experiment, the ascent rate of the UAV was set at a constant 3 m s$^{-1}$, and the maximum probing height reached approximately 1200 m above

ground level (AGL). The portable meteorological instruments recorded data at a sampling frequency of 1 Hz, with automatic data storage during the flights.

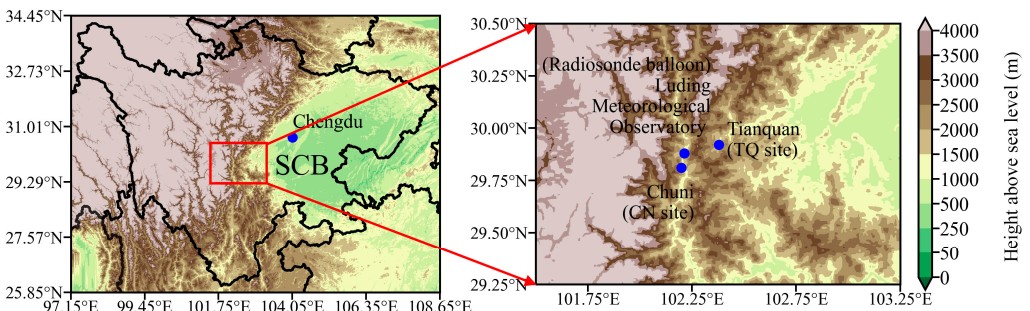

**Figure 1.** The topography of ESTP (**left**) indicating the locations of two UAV sites at CN and TQ and Luding Meteorological Observatory (**right**). The colored background shows terrain heights above sea level (m). This figure was plotted using equidistant cylindrical projection.

The UAV was scheduled to be launched every 4 h from 01:00 on 7 July to 21:00 on 16 July 2022, local standard time (LST) GMT+8. All flight times during the experiment are shown in Table S1. From 7 July at 01:00 to 11 July at 13:00, the weather conditions were mostly sunny, resulting in a total of 22 flights at the CN site. Among these flights, altitudes ranged from 650–700 m in 11 flights, 800 m in 2 flights, and 1200 m in 9 flights. From 11 July at 19:00 to 16 July at 21:00, observations were conducted at the TQ site under predominantly cloudy weather, with occasional rainfall leading to a higher number of missing data points. During this period, a total of 17 flights were conducted. Among these flights, 1 reached a maximum altitude of 700 m, 1 reached 1000 m, and 15 reached 1200 m. The regular launch intervals enabled comprehensive data collection throughout the observation period.

Above the UAV, we installed a GTS1-2 (Nanjing Daqiao Machinery Co., Ltd., Nanjing, China) to measure temperature (T, °C), relative humidity (RH, %), atmospheric pressure (P, hPa), and an FT205 (FT Technologies Limited, Sunbury, UK) to measure wind speed (WS, m s$^{-1}$) and direction (WD, °). Beneath the UAV, a portable optical particle spectrometer (POPS), manufactured by Handex in the United States, was mounted to measure the physical parameters of atmospheric aerosols. With its compact weight of less than 1 kg, the POPS device is particularly suitable for deployment on balloons and small unmanned aerial vehicles (UAVs). It utilizes Mie scattering, based on the principle of light scattering by individual particles within the size range of approximately 140–3000 nm, to measure the optical size of the sampled aerosol particles [58]. In this experiment, aerosol particles ranging in size from 130 nm to 3000 nm were classified into 16 size bins, and the sampling frequency was set at 1 Hz. For UAV data error specifications, refer to Table S2.

### 2.2.2. Radiosonde and Surface Observations

In addition, we launched 10 radiosonde balloons from Luding Meteorological Observatory, located at (102.21°N, 29.88°E, 1387 m) at the same time as the UAV (see Table S1). This station is in the western foothills of Erlang Mountain, which is in close proximity to the CN site. The radiosonde balloons were launched simultaneously with the UAV flights, enabling the collection of data on wind direction, wind speed, temperature, relative humidity, and atmospheric pressure. The radiosonde original measurements had a temporal resolution of about 1 s, an ascent speed of 400 m·min$^{-1}$, and a data accuracy of 0.1 °C. The average vertical resolution was about 6–7 m from the surface to the lower troposphere, which ensured a high-resolution description of the thermodynamic structure in ABL [59,60]. In this study, the original radiosonde and UAV data at each time step were subjected to three-fold spline interpolation and then uniformly processed at 5 m intervals to eliminate random errors. The measurements obtained from the radiosondes provide a representation of the atmospheric conditions throughout the observation period and serve to validate the accuracy of the UAV measurements.

As indicated in Figure 2, a comparison of the data obtained from the UAV and radiosonde balloons at the CN and TQ sites reveals the level of correlation between these measurements. The results demonstrate that the data collected from the radiosonde showed a stronger correlation with the UAV data at the CN site, which is situated in the western foothills of Erlang Mountain. The correlation coefficients of temperature and relative humidity were 0.8976 and 0.8189 (a chi-square test was applied with the *p*-value set as 0.01). Conversely, the correlation between the radiosonde data and the UAV data at the TQ site, located in the eastern foothills of Erlang Mountain, was comparatively weaker, particularly in relation to relative humidity. The correlation coefficients of temperature and relative humidity were 0.7145 and 0.2621 (a chi-square test was applied with the *p*-value set as 0.01). This discrepancy suggests that Erlang Mountain exerts a substantial influence on the blocking effect of water vapor transport. In addition, Luding Meteorological Observatory also provided meteorological data in accordance with the specifications for surface meteorological observation—automatic observation (GB/T 35237-2017),which were hourly average data including relative humidity (RH, %), temperature (T, °C), precipitation (mm, h), sea level pressure (P, hPa), wind, and sunshine duration (SD, h).

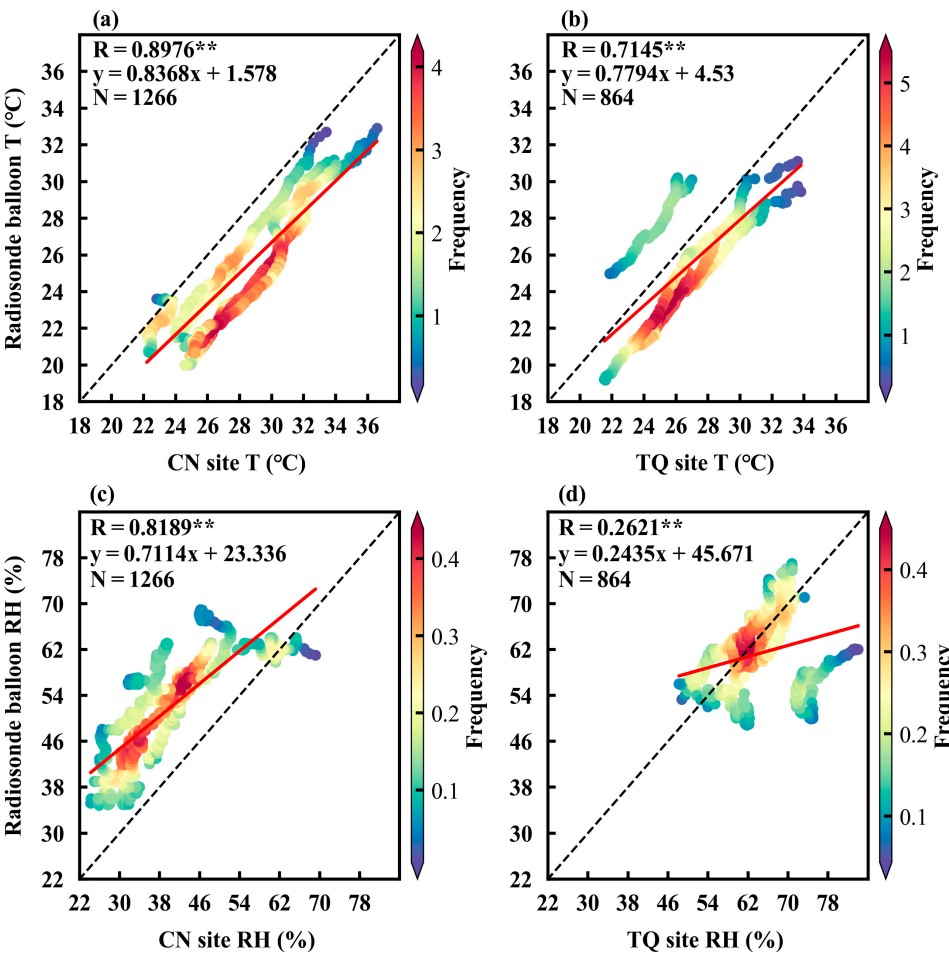

**Figure 2.** The temperature (T) and humidity (RH) scatter density of UAV and radiosonde balloon data at the same time, from the CN site (**a**,**c**) and the TQ site (**b**,**d**). ** signifies that the result has passed the significance test at the 0.01 level of confidence.

### 2.2.3. ERA-5 Data

A new generation of reanalysis data, the European Center for Medium-Range Weather Forecasts Reanalysis Version 5 (ERA-5) reanalysis data (https://www.ecmwf.int/, accessed on 13 February 2023) from the ECMWF [61], were also used in this study. The ERA-5 reanalysis data provide meteorological elements such as mean sea level pressure, relative

humidity, u-component of wind, v-component of wind, and vertical velocity with a spatial resolution of $0.25° \times 0.25°$. Hourly data from 7 July 2022 to 16 July 2022 were analyzed to examine the temporal and spatial variations of the atmospheric background field during the observation period.

### 2.3. Determination of PBLH

In this study, UAV data were employed to estimate the planetary boundary layer height (PBLH) based on the 1.5-theta-increase method [62]. According to this method, the PBLH is defined as the first altitude at which the potential temperature surpasses the minimum value within the PBL by 1.5 K. This methodology has been widely applied in previous studies investigating the PBL, demonstrating favorable accuracy and consistency in the obtained results [53,63,64]

### 2.4. Air Mass Sources

Previous research has indicated that variations in air masses can result in spatiotemporal discrepancies in the PNSD and aerosol optical properties [39,65,66]. In order to gain a deeper understanding of the sources of air masses and the pathways of aerosol transport within the study area, 72 h air mass back trajectories for air masses at 1 km were analyzed using the NOAA Hybrid Single-Particle Lagrangian Integrated Trajectory (HYSPLIT) model [67].

## 3. Results and Discussion

### 3.1. Temporal Variation of Surface Meteorological Variables

In order to gain a comprehensive understanding of the vertical distribution of PNSD and the influence of meteorological conditions on it in regions characterized by complex terrain, field campaigns were conducted at ETSP. The surface meteorological conditions observed at the Luding Meteorological Observatory, including temperature (T, °C), relative humidity (RH, %), precipitation (mm, h), surface atmospheric pressure (P, hPa), sunshine duration (SD, h), and wind, are shown in Figure 3. These observations provide a comprehensive overview of the meteorological conditions throughout the observation field campaigns. For detailed information regarding the meteorological conditions at the observation points, namely the CN site and TQ site, where the UAVs were deployed, please refer to the subsequent analysis at Part 3.3. During field campaign observations, the daily average temperature displayed a decreasing trend, with the maximum temperature declining from 35.2 °C on 7 July to 28.2 °C on 16 July. However, owing to the impact of sunshine hours (characterized by clear skies and no precipitation), there was a temporary rebound in temperature on 13 and 14 July. Under favorable radiation conditions, we observed explosive growth of fine particles events (indicated by the gray shade at Figure 3) at both the CN and TQ sites. RH was higher than 85% on 11 July, which was conducive to hygroscopic growth of particulate matters. From 8 July, the intensity of the low-pressure system in the SCB increased, the low-pressure range expanded, and an obvious low-vortex circulation gradually formed at 850 hPa (Figure 4), causing the surface atmospheric pressure at Luding meteorological observatory to decrease from 853.6 hPa on 7 July to 843 hPa on 11 July. Under the influence of the low-pressure system, the southerly wind intensified, bringing in more water vapor and potentially leading to an increase in cloud cover [68], which corresponded to a decrease in sunshine duration during the later phase of the observation field campaigns, accompanied by multiple instances of precipitation. Therefore, at the TQ site, we observed hygroscopic growth of particulate matter (indicated by the green shade in Figure 3) and the precipitation clearance effect on particulate matter (indicated by the blue shade in Figure 3).

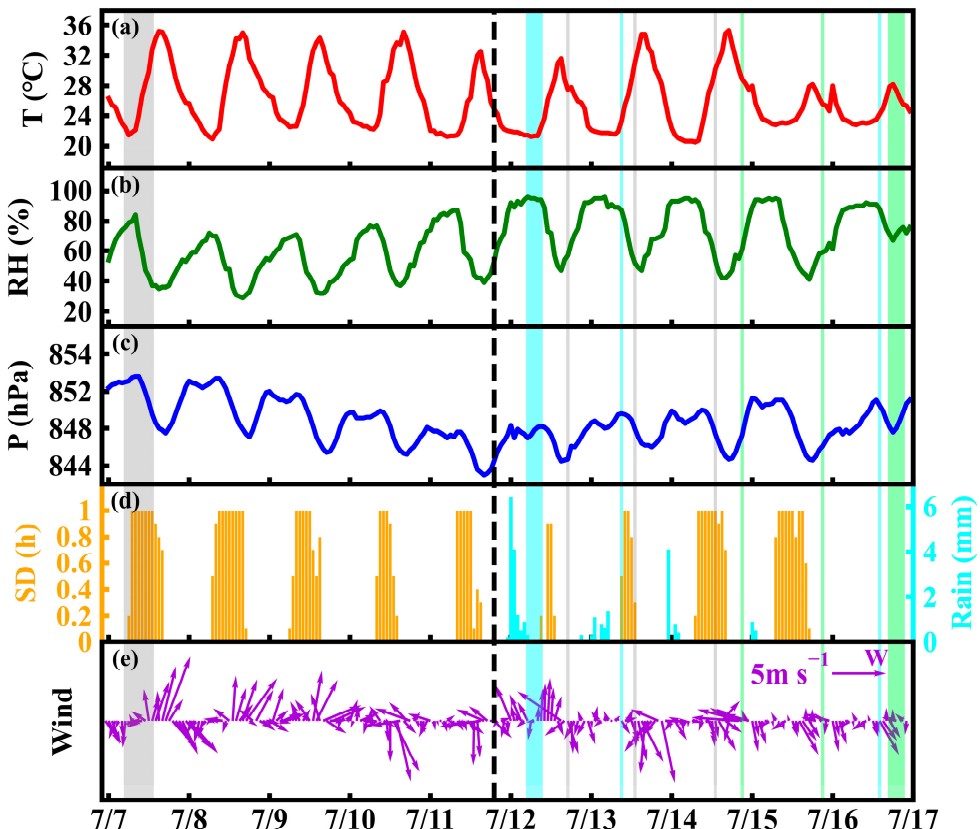

**Figure 3.** Temporal variations of (**a**) air temperature (T, °C), (**b**) relative humidity (RH, %), (**c**) surface atmospheric pressure (P, hPa), (**d**) sunshine duration (SD, h) and surface precipitation (mm, h), and (**e**) wind vector observed at Luding meteorological observatory from 7 July to 16 July 2022. The shaded gray, green, and blue parts represent the explosive growth of fine particles, the hygroscopic growth of particulate matters, and the precipitation clearance effect of particulates. The section to the left of the black dotted line represents the observation period at the CN site, while the section to the right of the black dotted line represents the observation period at the TQ site.

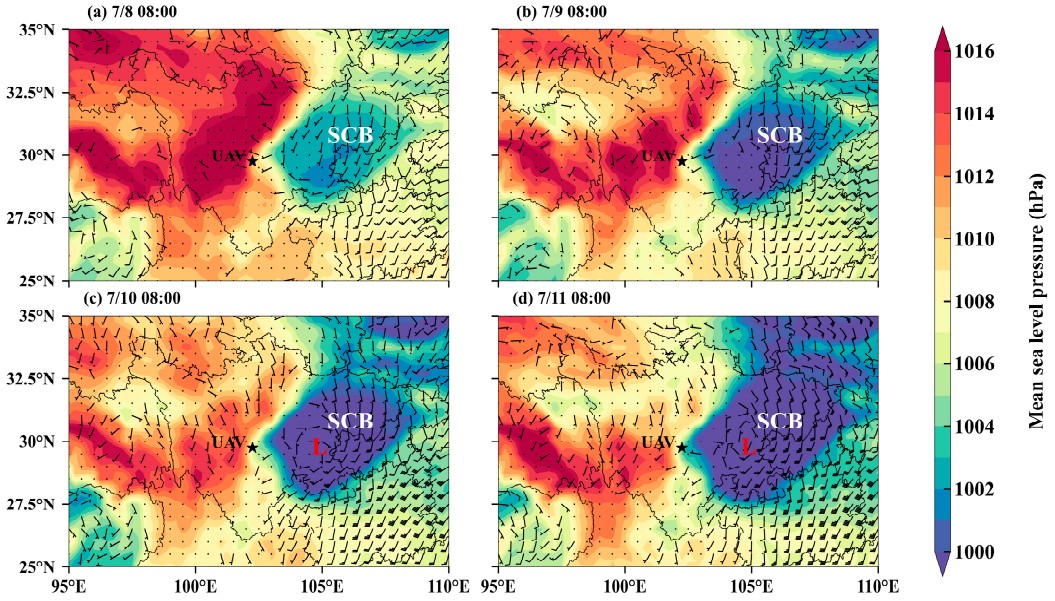

**Figure 4.** 850 hPa wind and mean sea level pressure from 8 July to 11 July at 08:00 LST.

### 3.2. The Relationship between Particle Number Concentration and PBLH Based on UAV Measurements

The PBL plays a crucial role in influencing the dispersion, transport, and transformation of aerosol particles in the atmosphere, and can affect the vertical distribution of PNSD. Therefore, understanding the intricate influencing of the diurnal variation of the PBL and aerosols in the vertical direction is a crucial aspect when investigating the impact of the PBL on the vertical distribution of PNSD. Therefore, the PNC, temperature, wind, and RH data with a vertical resolution of 3 m s$^{-1}$ were collected by the UAVs through three spline interpolations to eliminate random errors, and profiles with 5 m for each layer's vertical resolution were obtained. Subsequently, these data were stratified at 50 m intervals, and the average value of each group was calculated to draw the vertical profiles.

The averaged diurnal evolutions of the vertical distributions during the observation period shown in Figure 5 represent the accumulation and diffusion process of pollutants at different observation sites with the change of PBLH. First, the PBLH plays a crucial role in the dispersion and vertical distribution of particles. As the diurnal variation of the PBLH occurred, a corresponding synchronous change was observed in the total PNC at the PBLH. As shown in Figure 5, during the observation period, it was generally observed that the height of the daytime PBL exceeded that of the nighttime PBL. After sunrise (05:00–09:00 LST), the increase of surface radiation flux fosters an augmentation in mixing processes within the PBL, thereby promoting a robust development of the PBLH. At the CN site, the PBLH increased by an average of 204 m, and the particle number concentration decreased by 16.43%. At the TQ site, the PBLH increased by an average of 86.6 m, and the particle number concentration decreased by 58.76%. The significant decrease in PNC at the TQ site can be attributed to two possible factors: first, the dispersion facilitated by the uplift of the PBL, as depicted in Figure 5; and second, the precipitation removal, as demonstrated in Figure 3, which has been previously discussed [69,70]. Throughout the afternoon and evening, specifically from 13:00 to 21:00 LST, the PNC at the top of the PBL demonstrated a continuous increase due to the decreased PBLH. However, from 01:00–05:00 LST at the CN site and 09:00–13:00 LST at the TQ site, the PNC did not exhibit a decrease despite the increasing PBLH. This phenomenon could be attributed to two potential factors: the accumulation of particles caused by a temperature inversion (Figure S1), and the regional nucleation process [71].

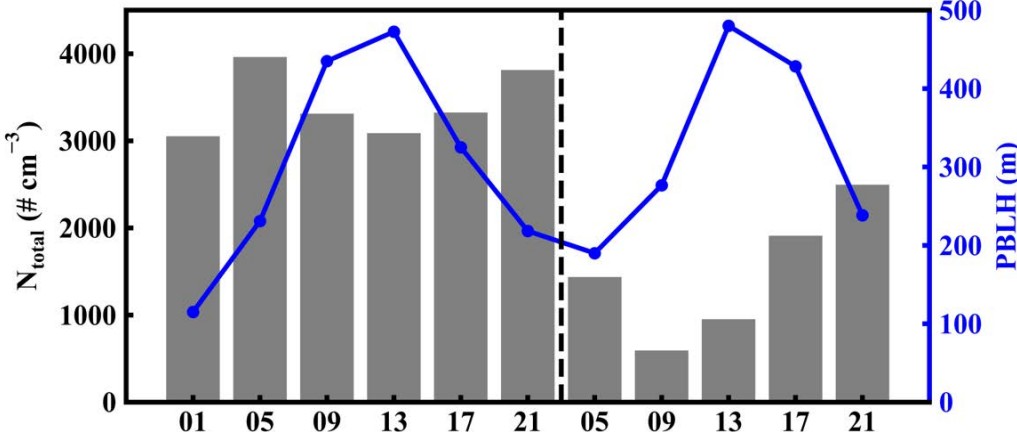

**Figure 5.** Averaged diurnal variation of the PBLH and the total PNC (N$_{total}$) at the PBLH (gray histogram) from 01:00–21:00 LST at the CN site (left of the dotted line) and 05:00–21:00 LST at the TQ site (right of the dotted line).

Figure 6 depicts a typical process in which daily changes in PBLH affected the total PNC. From 01:00–05:00 LST on 8 July, the area below 200 m AGL experienced near-isothermal conditions. During this period, the total PNC below 200 m reached approximately 3600–4000 # cm$^{-3}$. After sunrise, as the temperature increased, the stable boundary layer dissipated, and the PBL rapidly expanded. By 13:00, the RH within the PBL was below

32%, and the PBLH reached 430 m. At this time, a three-layer structure can be identified in the vertical profile of the particles at 13:00 LST on 8 July (Figure 6a). From bottom to top, this structure comprises a mixed layer with a uniform PNC, a transition layer where PNC decreases with height, and a low PNC in the free atmosphere, similar to the vertical distribution of pollutants in a previous study [53]. Subsequently, from 17:00–21:00, as the temperature decreased and the PBLH approached 220 m, the total PNC began to rise.

Furthermore, during the nighttime, we observed temperature inversion wherein particles were trapped near the surface and facilitated the accumulation of particles (Figure S1).

In conclusion, the structure of the PBL plays a pivotal role in governing the dispersion and vertical distribution of PNC. However, particle microphysical effects should also be considered, including NPF and hygroscopic growth, as these factors are critical in comprehending PNSD.

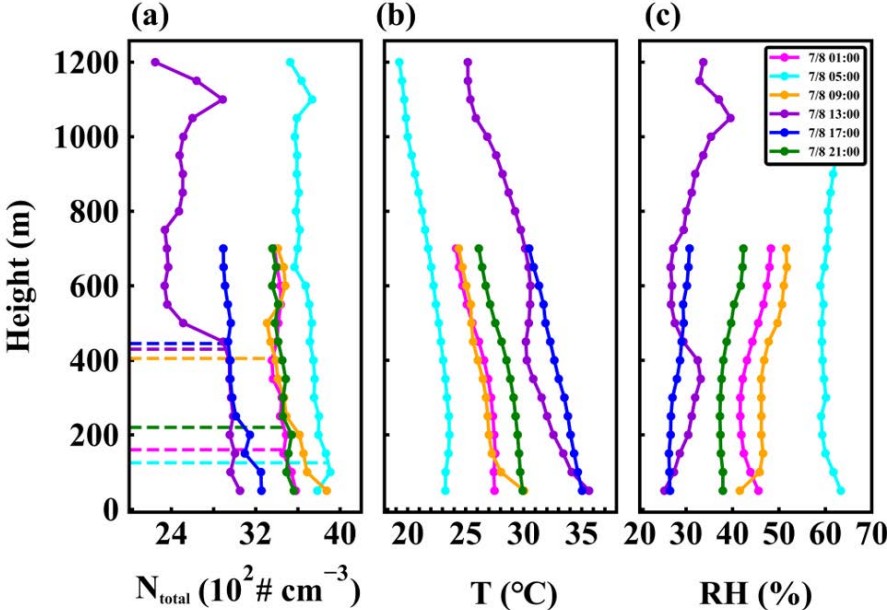

**Figure 6.** The UAV data from 8 July 2022, showing diurnal variation of vertical profiles of the total PNC (**a**), temperature (**b**), relative humidity (**c**) and PBLH (dashed lines in Figure 6a).

### 3.3. Case Studies of Particle Number Concentration Changes under Various Conditions

The evolution of particles at small particle sizes is another critical aspect to consider affecting PNSD. Small particles can undergo various physicochemical mechanisms, such as particle nucleation, condensation, agglomeration, aging, hygroscopic growth, activation as cloud condensation nuclei (CCN), etc., leading to a change in particle sizes [4]. These processes significantly impact the PNSD and hold implications for atmospheric phenomena like cloud formation and air quality. Understanding the dynamics of small particle changing is essential for accurately characterizing particle populations in the atmosphere [72–74]. In the subsequent sections, we will present three cases that have influenced the vertical profiles of PNSD during the observation period.

#### 3.3.1. Explosive Growth of Fine Particles

At both the CN and TQ sites, we observed a total of four instances of rapid fine particle growth, specifically in the particle sizes range of 130–272 nm (Figure S2). In this section, we present two representative events of fine particle growth, one at the CN site and the other at the TQ site. For clarity, we denote the number concentration of particle sizes range of 130–156 nm as $N_{130-156}$, the number concentration of particle sizes in the range of 226–272 nm as $N_{226-272}$. Figure 7a illustrates the vertical profile of PNSD (at left, the color indicates different particle size ranges), wind (at right, colored dots represent the wind direction), and RH (black line) at the CN site at 5:00 LST on July 7. Within the

layer between 100 m and 350 m AGL, the RH ranged from 67% to 70%. Combined with the data presented in Figure S1, the temperature inversion played a significant role in the accumulation of fine particles in the lower layer. Within this layer, the PNC increase to a maximum of $N_{130-156}$–$N_{226-272}$ was 90.82 # cm$^{-3}$. Additionally, a wind shear was observed between southerly and northerly winds at 500 m, with the $N_{130-156}$–$N_{226-272}$ value reaching 89.72 # cm$^{-3}$. Similarly, wind shear was observed between easterly and northerly winds at 650 m, with a $N_{130-156}$–$N_{226-272}$ value of 71.55 # cm$^{-3}$. Wind shear can enhance turbulence, promote particle collision and aggregation, and increase the likelihood of particle nucleation and growth processes, and lead to the occurrence of NPF, ultimately increasing the concentration of fine particles [75].

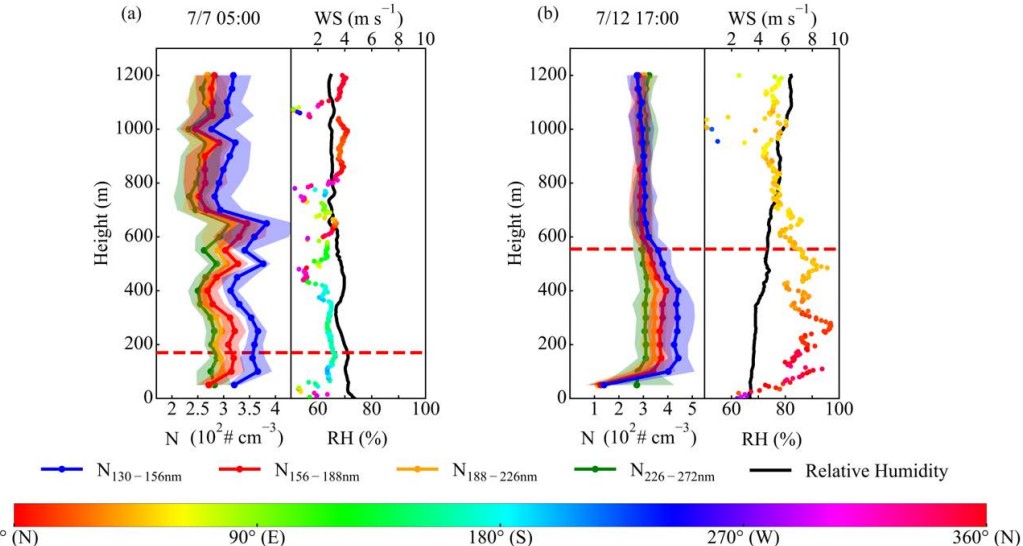

**Figure 7.** (**a**,**b**) Vertical profiles of PNSD (particle sizes ranged from 130 nm to 272 nm), wind and relative humidity compiled from UAV data. PNSD is averaged in the intervals of 50 m in altitude. Horizontal error shadows of PNSD represent standard deviations of PNSD at altitude intervals of 50 m. The red dotted line represents the PBLH.

In Figure 7b, another interesting phenomenon is shown at 17:00 LST on 12 July, where we observed a significant increase in the $N_{130-272}$, particularly in the $N_{130-156}$, within the PBL characterized by RH ranging between 67% and 73%. The maximum concentration of $N_{130-156}$–$N_{226-272}$ was 132.36 # cm$^{-3}$. Furthermore, the prevailing wind direction was northerly, as indicated by previous studies, which was known to favor NPF events [14,39]. Remarkably, the phenomenon of explosive growth of fine particles was confined to the PBL at this time. This phenomenon may be attributed to the sufficient vertical mixing within the PBL during the afternoon, which created regions of intense turbulence, further facilitating the formation of NPF events. The combination of favorable wind shear conditions and vertical turbulence mixing within the PBL contributes to the phenomenon of particle growth in specific regions and times [75].

While the direct observation of particles in the size range supporting NPF was not achieved, we did indeed observe an increase in the PNC within the diameter range of 130 nm to 272 nm under conditions favorable for NPF, including low humidity and wind shear. Consequently, it is inferred that NPF constitutes a possible contributing factor to the observed growth of fine particles within the size range.

3.3.2. The Process of the Hygroscopic Growth of Particles in the Background of Precipitation Clearance Effect

At the TQ site, it is noted that the RH was high due to rainy weather, as shown in Figure 3. During the period from 21:00 LST on July 14 to 21:00 LST on July 16, there were three instances when the RH exceeded 85%, and the PNC with particle sizes range from

130 nm to 272 nm ($N_{130–272}$) decreased, while the PNC with particle sizes ranging from 272 nm to 570 nm ($N_{272–570}$) remained stable or increased. In this section, we present an example of the hygroscopic growth of particles in a clean atmosphere after precipitation cleared (additional instances are as shown in Figure S3).

As depicted in Figure 8a, at 14:00 LST on July 16, a significant decrease in PNC within the PBL was due to the washing effect of precipitation. The air became notably cleaner, and the PNC above the PBL approached zero. However, at 17:00 LST, the total PNC started to accumulate again (Figures 8b and S4). In specific layers below 200 m AGL and between 400 m and 600 m AGL, a slight decrease in the $N_{130–272}$ was observed, while the $N_{272–570}$ exhibited a slight increase. This trend was particularly noticeable in the layers of 50–150 m, 200–350 m, 400–550 m, and 650–750 m AGL at 21:00 LST on July 16, as shown in Figure 8c. Below 800 m AGL, the maximum decrease rate of $N_{130–272}$ in the vertical direction was approximately 1653 # $cm^{-3}$ $km^{-1}$, while the maximum increase rate of $N_{272–570}$ in the vertical direction was around 3098 # $cm^{-3}$ $km^{-1}$. Given that the RH during this period exceeded 85% within the PBL, it could be hypothesized that the observed phenomenon could be attributed to the hygroscopic growth of particles. It is likely that particles with diameters below 272 nm underwent hygroscopic growth, leading to an increase in particle sizes and shifting to the range below 570 nm. Moreover, in the atmosphere above the PBL, where RH mostly exceeded 90%, the PNSD information observed at this altitude was very likely to reflect the situation of CCN in the cloud [76]. Thus, the increase in $N_{272–570}$ could also be attributed to the contribution of fine particles becoming activated as CCN. According to Köhler's theory, larger particles have a higher probability of being activated as CCN. Consequently, aerosol particles in the accumulation mode are more easily activated as CCNs, influencing the microphysical processes of clouds.

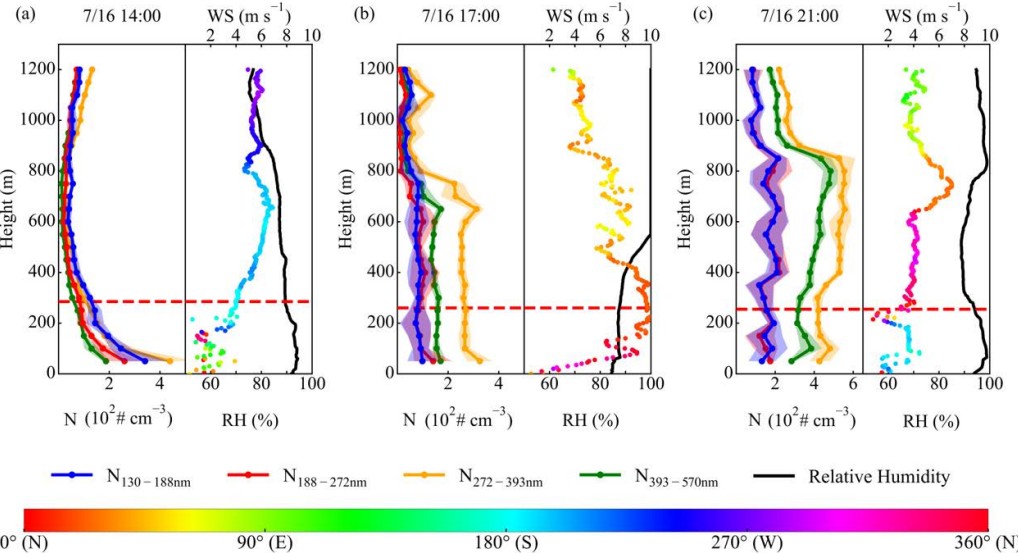

**Figure 8.** (**a–c**) Similar to Figure 7, but particle sizes range from 130 nm to 570 nm.

### 3.3.3. Influence of Air Masses on PNSD Profiles

Throughout the observation period, $N_{130–272}$ and $N_{272–570}$ exhibited multiple instances of continuous increase (Figure S4). To further investigate the air mass sources and aerosol transport pathways over the measurement area, 72 h air mass back trajectories for air masses at 1 km were analyzed using the NOAA Hybrid Single-Particle Lagrangian Integrated Trajectory (HYSPLIT) model [67]. The analysis of Figure 9a,b provides insights into the air mass sources influencing the CN and TQ sites. At the CN site, approximately 70% of the air mass passed through the SCB, which is densely populated and highly polluted; before reaching the observation site, 21.67% of the air mass came from Panzhihua City in the southwest of Sichuan Province, and another 8.33% originated from Aba (Ngawa) Tibetan and Qiang Autonomous Prefecture in Sichuan Province (Figure 9a). Similarly, at the TQ

site (Figure 9b), about 76.09% of the air mass passed through cities with frequent human activities in the SCB before arriving at the TQ site, and 23.91% of the air mass originated from Panzhihua region. Based on the findings that most of the air masses affecting the observation area originated from the SCB. As a typical representative city in the SCB, Chengdu experiences a significant presence of 250–500 nm particles, which accounts for 97.75% of the total concentration [77]. Based on this observation, it is reasonable to infer that the higher value of $N_{272-570}$, compared to other particles, during the observation period could be attributed to aerosol transport from the SCB region.

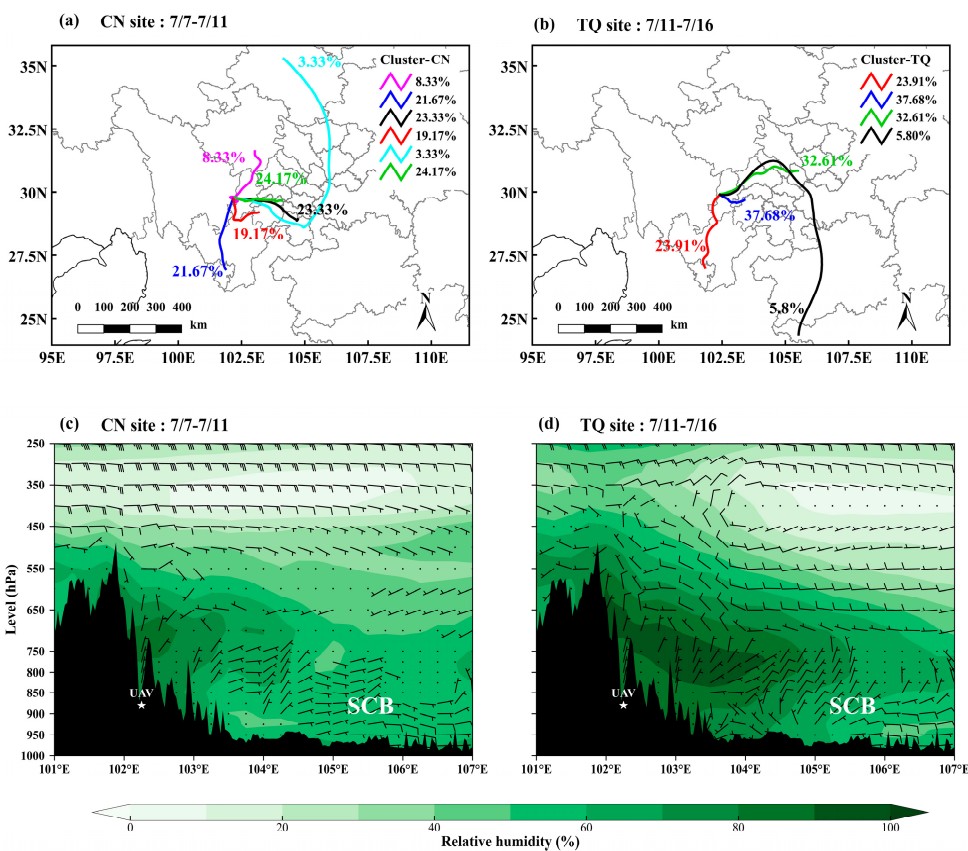

**Figure 9.** Cluster analysis of 72 h HYSPLIT back trajectories at an altitude of 1000 m from the CN site (**a**) and TQ site (**b**). The color of trajectories indicates different air mass sources and averaged vertical profiles of wind and relative humidity (section along 29.75°N) at the CN site (**c**) and TQ site (**d**). Black shade indicates terrain.

From 8 July, the surface low-pressure center in the SCB strengthened significantly, and the area of low pressure expanded to cover the observation area. Additionally, a prominent low-vortex circulation gradually formed at 850 hPa (Figure 4), which facilitated more uniform mixing of polluted air masses from the SCB. Furthermore, the geographical characteristics of the Tibetan Plateau played a role in the transport of air masses to the observation area in the ESTP, leading to a blocking effect. Consequently, a downdraft was generated over the observation sites, hindering the transport of water vapor, particulate matter, and other substances across the plateau (Figure 9c,d). Consequently, these substances accumulated within the ESTP, leading to the observed accumulation of aerosols at the observation sites. Furthermore, the intricate valley terrain surrounding the observation sites imposed constraints on the dispersion of aerosols to other regions.

This part analysis underscores the impact of regional transport and complex terrain on the PNC observed at the CN and TQ sites. Such influence is shaped by the combination of atmospheric dynamics, complex topography, and local meteorological conditions within the ESTP.

## 4. Conclusions and Discussion

This study, the first vertical study of PNSD from 7 to 16 July 2022 on the ESTP, was undertaken using UAVs equipped with POPS and meteorological sensors. The vertical distribution of aerosols and meteorological data was analyzed to reveal the variation characteristics in the vertical profile of PNSD in the ESTP region during the formation period of the Sichuan Basin vortex, thereby shedding new light on the distribution characteristics of accumulation mode aerosols in the intricate topography of the ESTP.

The observations revealed a significant negative correlation between the PBLH and the PNC, and the correlation coefficient was $-0.19$. During the morning, the rise in the PBLH at the CN and TQ sites caused decreases of 16.43% and 58.76%, respectively, in the PNC. Three distinct profiles were classified. In Type I, the explosive growth of fine particles within the size range of 130 nm to 272 nm could be attributed to the occurrence of NPF events under conditions of low humidity, strong wind shear, and northerly winds. The significant increase in particle concentration during such events can result in a maximum difference between $N_{130-156}$ and $N_{226-272}$ reaching 132.36 # cm$^{-3}$. Type II particles with a size range of 130–272 nm hygroscopic growth into larger particles (e.g., 226–272 nm) under high humidity conditions (RH > 85%), with a maximum vertical change rate of about $-1653$ # cm$^{-3}$ km$^{-1}$ for $N_{130-272}$ and about 3098 # cm$^{-3}$ km$^{-1}$ for $N_{272-570}$. The enlarged particles are likely to be activated as CCN within the cloud and affect the precipitation process. The decrease in $N_{130-272}$ and the stabilization or increase in $N_{272-570}$ could be attributed to multiple factors, including enhanced hygroscopic growth of particles, condensation, agglomeration processes, and possible CCN activation. Type III show the impact of the regional transport from the SCB and meteorological conditions on the PNSD with the ESTP region. During the occurrence of a surface low-pressure center and an 850 hPa low-vortex circulation in the SCB, polluting air masses originating from urban agglomeration are transported to the ESTP region, resulting in an observed increase in the PNC below 600 nm. Overall, this study sheds light on the complex various factors affecting the vertical profiles of PNSD in the ESTP region, including regional transport, meteorological conditions, and particle growth processes, which will help us further understand the distribution of aerosols or CCNs in this region. Moreover, it offers valuable insights into the mechanisms underlying the formation of heavy precipitation events in both local and downstream areas (such as the Sichuan Basin).

**Supplementary Materials:** The following supporting information can be downloaded at: https://www.mdpi.com/article/10.3390/rs15225363/s1.

**Author Contributions:** H.W. proposed and guided the study, provided the ideas, financial support, and revised the manuscript; C.S. processed, visualized, and analyzed data, and drafted the original manuscript with contributions from all authors; L.Z. completed the field observation and pre-data quality control; Y.Y. processed, visualized, analyzed data, and revised the manuscript; X.Z. and X.J. refined the design of the field observation and provided valuable advice. M.L. provided financial support; H.H. and D.P. assisted observation. All authors have read and agreed to the published version of the manuscript.

**Funding:** This work was funded by the National Natural Science Foundation of China (NSFC) research project (Grant No. 42105073), Heavy Rain and Drought-Flood Disasters in Plateau and Basin Key Laboratory of Sichuan Province (Grant No. SZKT202102), the Project of the Sichuan Department of Science and Technology (Grant No. 2022NSFSC1074).

**Data Availability Statement:** Measurement data described in this manuscript can be accessed at the data repository maintained by Mendeley Data. V1, doi: 10.17632/dvjvjhbjtf.1, https://data.mendeley.com/datasets/dvjvjhbjtf/1 (accessed on 31 July 2023) [78].

**Acknowledgments:** We greatly appreciate the contributions of three reviewers that helped improve the quality of the manuscript. Thanks also to Jiang Mengjiao for her guidance. We thank all participants of the Eastern Slope of the Tibetan Plateau field campaign for their cooperation.

**Conflicts of Interest:** The authors declare no conflict of interest.

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
