# Peer review of "Vertical Profiles of Particle Number Size Distribution and Variation Characteristics at the Eastern Slope of the Tibetan Plateau"

_remotesensing, doi:10.3390/rs15225363_

Round 1

Reviewer 1 Report

Comments and Suggestions for Authors

The authors perform UAV vertical profile measurements of the particle number size distribution (PNSD), relative humidity, pressure, temperature, sunshine duration wind speed/direction in two regions located on the Eastern Slope of the Tibetan Plateau and found (i) negative correlations between the PBLH and the particle number concentration (PNC), (ii) an explosive growth rate in fine particles in low humidity and strong wind shear, (iii) hygroscopic growth in larger particles in high humidity conditions and (iv) an increase in the PNC below 600 nm that results from polluting air masses.  Their study highlights conditions that affect the vertical profiles of the PNSD in this region and elucidates factors that impact aerosol variations related to the physical character of the atmosphere.

Overall, the paper is fairly well-written (some minor changes are suggested below). The figures are of high quality.  The significance of the work appears to be drawing correlations between particle size/concentration and the physical changes in the atmosphere that possibly lead to these changes.  I am not an expert in aerosol formation and growth processes, so it is difficult to assess the quality of their general conclusions about these two regions in the Tibetan Plateau. I feel these results will of interest to a narrow segment of the Remote Sensing community but within the scope of the journals focus.  I offer the following comments to be considered before publication.

1. The authors should clarify the meaning of the y-axis shown as dN/dlogDp in the Figure 5-8 captions and once somewhere in the text.

Page 1, Line 17: “platform obtained the” – remove “was”

Page 1, Line 22: “and the correlation coefficient”

Page 1, Line 23: “Three distinct profile characteristics have been”

Page 1, Line 29: Define SCB – Sichuan Basin

Page 3, Line 141: No cap on “influencing”

Page 7, Line 252: Seems like a typo “on the 1.5-?-increase method” – please clarify

Page 9, Line 310: No cap on “impact”

Comments on the Quality of English Language

The quality of English is fairly high.  A few typographical errors are noted above.

Reviewer 2 Report

Comments and Suggestions for Authors

The manuscript "Vertical profiles of particle number size distribution and variation characteristics at the Eastern Slope of the Tibetan Plateau" mainly focuses on the vertical distribution of aerosol PNSD observed by UAV and meteorological data, which reveals the variation characteristics for vertical profile of PNSD in the ESTP region during the formation period of the Sichuan Basin vortex, thereby shedding new light on the distribution characteristics of accumulation mode aerosols in the intricate topography of the ESTP. In general, the paper is well written and presented in a logical way. I therefore recommend publication of this paper in Remote Sensing after minor revisions. My comments are listed as follows:

1.      Line 17: the grammar of the first sentence should be revised in abstract.

2.      Line 29: Write the full name of the acronym when it first appears in the article.

3.      In introduction section: the first and the second paragraph should be rearranged. For example, the second paragraph would be added before the last sentence in the first paragraph.

4.      Line 138-141: When talking about UAVs in regions characterized by intricate underlying surfaces examine the vertical profiles of particle number-size distributions under clean conditions. The authors could cite other references, e.g. “Investigation of the Vertical Distribution Characteristics and Microphysical Properties of Summer Mineral Dust Masses over the Taklimakan Desert Using an Unmanned Aerial Vehicle”.

5.      Line 142-143: It is necessary to add the corresponding literature on the time resolution of other vertical observation methods and explain what the time resolution is using UAV observation.

6.      Line 191-194: Please add the name of the observation instrument and the manufacturer.

7.      Line 297: what is the black dotted line in Figure 3?

It is suggested that it may be more intuitive to use a line chart for the PBLH in Figure 5.

Comments on the Quality of English Language

Minor editing of English language required

Reviewer 3 Report

Comments and Suggestions for Authors

This work used ten-days UAV observations of fine particles at two sites at the Eastern Slope of the Tibetan Plateau to analyze the vertical distributions of particle number and two effects of explosive growth and hygroscopic growth of fine particles. Unfortunately, some obvious deficiencies appeared in this work. The biggest weakness is lacking of convincing analysis. Some personal advices are suggested below.

1)     In the Instrument and data part, it lacks the error specification for the UAV observation, which is very important for in situ observation.

2)     For Fig. 2, was each point matched at the same time and same altitude? How to match?

3)     In Line 336-337, it says “This phenomenon could be attributed to two potential factors: the occurrence of New Particle Formation (NPF), and the regional transport of particles”. How to prove the two factors appeared?

4)     For Fig.6, the inversion appeared at 1:00 of July 7th from Fig. S1, not on July 8th. Then, the explanation in Line350-352 is not suitable. Additionally, the vertical axis scales in subplots of Fig. S1 are not consistent.

5)     The analysis for Fig.6 is superficial. It is better to give the profiles of T and Rh, and the backward trajectory. Also, the introduction of Fig. 6b in the caption is not matched.

6)     In Fig. 7b, what’s the reason for the fast decrease of particle number concentration below 150m?

7)     For section 3.3.1, what’s the definition of rapid fine particle growth? Change spatially or temporally? Did the author consider the naturally decrease of particle size with the altitude?

8)     According to the author’s explanation in Line 390-395, the wind shear induces the fast increase of particles. How to explain the similar large difference at the altitude of 200m, where there is no wind shear?

9)      In Line 412, it mentioned “high temperature”, but no temperature profile is found.

10)  From a to b in Fig.8, it was found that the concentration of N272-393 was 2-3 times of that of the others. What’s the reason for this phenomenon?

11)  The analysis of Fig. 9 does not combine well with the previous results.

Round 2

Reviewer 3 Report

Comments and Suggestions for Authors

For Fig. 6a, it shows profiles of total PNC (means a integrel from a size distribution?), I suggest to use N as the caption for X axis. Also for the other similar figures.

Author Response

Please see the file attached:
